# Research Progress of Coating Preparation on Light Alloys in Aviation Field: A Review

**DOI:** 10.3390/ma15238535

**Published:** 2022-11-30

**Authors:** Nan Li, Qiang Wang, Fang Dong, Xin Liu, Peng Han, Yu Han

**Affiliations:** 1School of Metallurgical Engineering, Xi’an University of Architecture and Technology, Xi’an 710055, China; 2Experiment Testing Institute, Xinjiang Oilfield Company, Karamay 834000, China; 3Key Laboratory of Special Equipment Safety and Energy Saving for State Market Regulation, China Special Equipment Inspection and Research Institute (CSEI), Beijing 100029, China

**Keywords:** cold spray, supersonic laser deposition, additive manufacturing, coating, repair, aviation

## Abstract

This paper systematically introduces the application status of coating-preparation technology on light alloys in the field of aviation parts repair. Included are the advantages and disadvantages of thermal spraying technology and laser cladding technology in the application process, as well as the research status and application prospects of the emerging cold spray (CS) technology and supersonic laser deposition (SLD) technology. Compared with traditional thermal-spraying technology, CS has many advantages, such as low spraying temperature, low oxygen content of the coating, and low porosity, which can effectively avoid oxidation, burning loss, phase change, and grain length during thermal spraying. CS can prepare oxygen-sensitive, heat-sensitive, amorphous, and nanomaterial coatings that are difficult to prepare by traditional thermal-spraying technology. However, in the preparation of high-strength super-hard alloys, CS has shortcomings such as low deposition efficiency and bonding strength. SLD overcomes the shortcomings of CS while inheriting the advantages of CS. In the future, both technologies will be widely used in repairing and remanufacturing in the field of aviation. Based on the principles of CS and SLD, this paper introduces, in detail, the deposition mechanism of the coating, and the specific application examples of CS in the aviation field at the present stage are described. The research and application status of the two technologies in the fields of anti-corrosion coating, wear-resistant coating, functional coating, repair, and remanufacturing in recent years are reviewed. Finally, the application and development prospects of CS and SLD are discussed.

## 1. Introduction

In recent years, with the rapid development of the aviation industry, lightweight design is a necessary guarantee to achieving low carbon emission and long-range flight. Light-metal alloys, such as aluminum (Al), magnesium (Mg), and titanium (Ti) alloys, have the advantages of low density and high specific strength [1,2,3,4,5,6]. In addition, Mg alloys have good electromagnetic shielding and high damping characteristics and have been widely used in carrier-based helicopters and fighter jets [7,8]. Due to their high strength and manufacturing feasibility, Al alloys are also widely used in civil aviation. As shown in Table 1, Al alloys occupy a critical position in aircraft manufacturing [9]. Figure 1 shows the specific use of Al alloys in the current mainstream wide-body passenger aircraft Boeing 777 [9], including aircraft trusses, landing gear, aircraft skins, wing spars, and other structural parts as well as non-stressed parts such as seat rails and floors. Ti alloys are commonly used in aircraft structural parts, engine casings, aircraft landing gear, flame frames, and other bearing components [10]. The tensile strength of Ti6Al4V alloys commonly used in aircraft structures is nearly 1 GPa [11].

During the service period, the surface damage of light alloys caused by corrosion and wear seriously affects the aircraft’s flight safety and service life [12,13,14]. Due to the high value of these parts, the manufacturers generally do not store spare parts. Therefore, repairing the damaged parts is the most efficient way to save maintenance time and cost, producing huge green and circular-economic benefits [15].

## 2. Preparation of Coatings on the Surface of Aviation Parts

At present, the technologies for preparing alloy coatings on metallic surfaces mainly include thermal spraying technologies, such as high-velocity oxygen fuel spraying (HVOF) [16,17], arc spraying [18,19], plasma spraying [20,21,22], and laser cladding (LC) [23,24]. The prepared coatings mainly include Fe-based, Cu-based, and Al-based coatings, and are mainly used for preparation of wear-resistant and anti-corrosion coatings, as seen in Table 2. In the thermal spraying process, the alloy powder particles are heated to a molten or semi-molten state and then rapidly solidify to form a coating after being deposited on the surface of the substrate within a specific speed range. However, the rapid heating and cooling involved in the thermal spraying process can easily lead to problems of oxidation, phase transformation, compositional segregation, and the generation of shrinkage cavities, so the microstructure of sprayed coatings is relatively complex, usually containing oxide inclusions, un-melted particles, and pores that adversely affect the performance of the coating. For example, Wang et al. [25] used HVOF technology to prepare a Fe-based alloy coating on the surface of Al alloy, and the analysis results showed that the coating had higher hardness and better wear performance than the Al alloy substrate, but there were transparent oxides and pores of different sizes on the particle interface in the coating. The larger pores were caused by molten particles entrapped in the air during flight, and small-sized pores were caused by shrinkage during rapid solidification. Zhang et al. [26] studied the corrosion behavior of Fe-based alloy coatings prepared by HVOF in NaCl solution, and the results showed that the pitting phenomenon originated from the particle interface region in the coating. The reason for this phenomenon was that the oxidation during the spraying process led to the generation of the Cr element depletion area, and the potential difference between the Cr-depleted and Cr-rich areas caused corrosion. The production of oxides in the alloy coating destroys the compositional balance of the parent alloy and quickly leads to phase separation. In addition, high heat input will form residual tensile stress on the surface of the substrate, which can also cause severe component dilution at the coating–substrate interface. In addition, the residual tensile stress caused by thermal influence and phase transformation at the interface will cause the matrix to crack, forming obvious cracks. The initiation of cracks at the interface will significantly reduce the service life of repaired parts.

LC technology uses the characteristics of rapid heating and cooling of the laser source to prepare coatings [29]. The energy density of the laser is usually between 10^4^ and 10^6^ W/cm^2^, and the cooling rate of the molten pool can reach 10^4^–10^6^ K/s. The high energy density completely melts the surface of the cladding material and the substrate material, resulting in a skinny molten pool. The LC technology usually involves a large heat-affected zone (HAZ) on the substrate surface. At the same time, in the multi-pass cladding process, the repeated heating by the subsequent scanning quickly leads to grain growth of the deposited coating. Due to the vast heat input, there is a potential corrosion difference between the coating and the substrate, which leads to pitting corrosion at the interface, thereby reducing the corrosion resistance of the coating [27], For example, Katakam et al. [30] studied the corrosion behavior of LC Fe-based alloy coatings and showed the precipitation of carbide phases in the coatings, which reduced the corrosion resistance of the coatings. Li et al. [31] investigated the corrosion properties of LC Ni-based alloy coatings and compared them with strips of the same composition and 316L stainless steel. The results showed that in the same corrosion environment, the corrosion potential of the LC layer was lower than that of the strip and 316L stainless steel. The reason is that the phase interface contained in the LC layer provides an effective diffusion channel for charge transfer during the corrosion process, which accelerates the penetration of the corrosion solution, thereby reducing the corrosion resistance of the coating.

In summary, in preparing alloy coatings, although the structure and properties of the coating can be optimized by adjusting the process parameters, it is still difficult to avoid high-temperature conditions that lead to adverse effects. In addition, the thermal damage by thermal spraying and LC technologies to the substrate surface is severe. The grain growth and composition segregation in the heat-affected zone will adversely affect the interface bonding between the coating and the substrate, seriously affecting the service performance of the repaired parts. Therefore, it is of great practical significance and theoretical innovation value to seek a new preparation method of alloy coating to eliminate the negative impact of high-temperature influence.

## 3. Preparation of Protective Coatings by Cold-Spray Technology

In the mid-1980s, when the Soviet Union scientist Anatolii Papyrin and his team studied the model of two-phase flow (gas + solid particles) in a wind tunnel, they found that when the speed of the solid particles was higher than a certain critical value, the matrix material would no longer be eroded and particles began to adhere to the matrix [32]. Inspired by this phenomenon, Anatolii Papyrin and his team first proposed the concept of cold gas dynamic spraying (referred to as ‘cold spraying’) in 1990, which could be used as a new coating process. Later, they successfully deposited a variety of pure metals, metal alloys, and composites on different matrices, proving the possibility of cold-spraying technology in engineering applications [32]. Cold-spray (CS) technology is characterized by low-temperature solid-state deposition [33], and its process temperature is much lower than the melting temperature of the original powder material [34,35], which can effectively avoid the adverse effects of oxidation, phase transformation, and thermal cracking caused by high temperature [36,37,38]. It is very suitable for preparing anti-corrosion and wear-resistant coatings on the surface of Al, Mg, and Ti alloys which are oxidation-sensitive [39,40,41]. As shown in Figure 2, the technical principle of CS technology is to use high-pressure inert gas nitrogen (N_2_) or helium (He) as the energy source. The electric heater preheats the gas to a specific temperature to accelerate powder materials to a speed range of 500–1200 m/s through the supersonic nozzle. Particles undergo severe plastic deformation under high-speed impact to form coatings [34,42,43]. As shown in Figure 3, compared with the above-mentioned high-temperature coating-preparation technologies, such as HVOF, plasma spraying, and wire arc spray, CS has the characteristics of low process temperature and high spraying speed. The most important thing is that the particles will undergo significant plastic deformation during the deposition process compared with thermal spray coatings. Therefore, the coatings have extremely low porosity and high hardness, which are essential for improving the coatings’ wear and corrosion resistance [44,45,46]. In addition, CS is also suitable for surface treatment of matrix materials with low melting points and easy oxidation, such as Al alloys, Mg alloys, and Ti alloys [47]. Therefore, CS technology makes it possible to prepare high-performance coating materials on the surface of light alloy aircraft components.

In recent years, CS technology has been applied in the repair field of aircraft components, providing wear-resistant and corrosion-resistant coatings [48,49]. Mg is a very active electrochemical metal, and galvanic corrosion will occur when Mg alloys are combined with other structural metals in the presence of electrolytes or corrosive media [50,51]. The Al-based coatings prepared by CS technology have been applied to protect Mg alloy parts against corrosion. For example, Wei et al. [48] prepared dense Al and Al alloy protective coatings on the surface of AZ31B Mg alloys by CS technology. The experimental results showed that the CS coating provided long-term solid corrosion protection for the AZ31B Mg alloy substrate. The experimental results are shown in Figure 4. It can be seen that the protective coatings were still uniform and dense after 1000 h of corrosion experiments. Tao et al. [49] deposited pure Al coating on AZ91D Mg alloys by CS technology. Electrochemical tests showed that cold-sprayed pure Al coating has better pitting-corrosion resistance in neutral 3.5 wt.% NaCl solution than bulk Al of similar purity, as shown in Figure 5. It can be seen that the pitting-corrosion resistance of pure Al coating was better than that of bulk Al within ten days in a corrosive environment. This also proved that the cold-sprayed pure Al coating can be used as an anti-corrosion coating for the Al substrate. In the cold-sprayed Al alloy anti-corrosion protective coatings, Babu et al. [52] used CS technology to deposit aerosolized Al amorphous/nanocrystalline alloy powder on 6061 Al alloy substrates. As shown in Figure 6, the potentiodynamic polarization study showed that the thickness of the reaction layer formed during the salt spray corrosion tests was two to three times lower in the case of cold-sprayed Al amorphous/nanocrystalline coatings. The reaction-layer thickness of the heat-treated coating was further reduced. Pitting corrosion formed a porous oxide layer on the bare substrate surface, whereas uniform, dense, and thin corrosion layers were observed in sprayed and heat-treated coatings. Zhou et al. [53] deposited HAP/Ti composite coating on Ti substrate by CS technique to study the electrochemical corrosion behavior of cold-sprayed 20 wt.% and 50 wt.% HAP/Ti composite coating. The research results showed that the CS technique was suitable for the fabrication of HAP/Ti composite protective coating on Ti substrates.

In terms of wear-resistant coatings, the most common method for preparing wear-resistant coatings by CS technology is to spray a metal matrix composite coating, a mixture of hard particles and softer ductile particles. The softer binder phase ensures the opportunity efficiency and coating quality of the CS coating, while the hard wear-resistant particles ensure the hardness and wear performance of the coating [54,55,56]. For example, Chen et al. [57] studied the tribological properties of cold-sprayed aluminum/diamond composite coatings on pure Al substrate. The results showed that aluminum/diamond composites had better wear resistance than selective laser sintering Inconel 625 and 17-4PH alloys. Higher diamond content in the composite resulted in better wear resistance. The friction and wear experiment results are shown in Figure 7. Spencer et al. [58] prepared Al-Al_2_O_3_ and 6061Al-Al_2_O_3_ coatings on the AZ91E Mg alloy substrate by cold-spraying technology. The study showed that compared with the substrate, the wear rate of the composite coating was significantly reduced. In aviation maintenance, Goldbaum et al. [59] produced a bi-layer system using CS technology to improve the mechanical and frictional properties of repaired parts. This bi-layer system consisted of a cold-sprayed Ti coating and TiN, TiSiN, and TiSiCN HiPIMS protective coatings and demonstrated the use of the CS system for Ti friction-protective coatings.

## 4. Additive Manufacturing and Repair via Cold-Spray Technology

Additive manufacturing technology has changed the way products are manufactured and is a revolutionary breakthrough in the principles of manufacturing technology. Achieving the perfect integration of additive manufacturing and the overall configuration design of the structure is to give full play to the technological advantages of additive manufacturing. It is the key to breaking through the bottleneck of the traditional design mode and processing technology to reduce the structure’s weight further and improve its performance. The traditional additive manufacturing technology uses laser beams, electron beams, and arcs as heat sources to achieve the layer-by-layer formation of metal components by melting metal powders or wires. Depending on the energy source and the forming materials used, typical metal-additive manufacturing mainly includes selective laser melting (SLM) [60], electron beam melting (EBM) [61], electron beam freeform fabrication (EBFF) [61], and laser metal deposition (LMD) [62]. Since the processing temperature is higher than the material’s melting point, the traditional additive manufacturing process will cause problems, such as material oxidation, phase transition, decomposition, and grain growth, which significantly impact the final performance of the parts.

With the gradual improvement of the capabilities of CS equipment, this technology has now entered the field of 3D printing [39,63]. The US Army Research Laboratory (ARL) has an early research and application development of CS technology. In 2008, a corresponding technical standard (MIL-STD-3021) was proposed for the application of this technology in repairing damaged parts, marking that this technology had entered the stage of practical application in the field of military equipment [64]. As reported by ARL, this technology is currently widely used to repair Al alloy and Mg alloy damaged parts caused by corrosion and wear, such as the UH-60 helicopter gearbox (ZE41A, AZ91 Mg alloy) and fuel tank bottom (7075 Al alloy) [65,66,67,68]. Champagne et al. [69] used the CS process of 6061 Al alloy to join and provide a structural repair of cast ZE41A-T5 cast Mg. It demonstrated that the cold-spray process can produce dissimilar metal joints with comparable ultimate tensile strength, hardness, and shear strength to those of the repaired magnesium alloy. Cavaliere et al. [70] used 2198 and 7075 Al alloy powders to CS repair the 2099 aviation Al alloy panel with a 30° V notch on the surface. The results showed that with 2198 Al alloy repair, the crack resistance increased by more than seven times, and CS repair helped increase the cracked structure’s overall fatigue life. Cavaliere et al. [71] prepared Ti- and Ni-based nanocomposite coatings by CS technology for the repair study of Ti alloy substrates. The experimental results showed that CS could successfully deposit dense Ti coatings for repairing Ti alloy parts. Villafuerte et al. [72,73] reported the use of CS technology to repair aviation Al alloy components and for the on-site repair of aircraft landing gear, as shown in Figure 8.

In 2019, Australia’s Titomic company successfully joined CSIRO (Australian Commonwealth Scientific and Industrial Research Organization) to apply CS technology to manufacturing and developed a new 3D printing process—Kinetic Fusion. This process can print Ti alloys in the same way as other existing metal 3D printing processes, but with better performance, and the printing speed can reach 45 kg/h (about 10–100 times higher than traditional additive manufacturing technology). Compared with traditional laser 3D printing, the tensile strength of the finished product printed by this process was increased by 34%. In 2018, Titomic agreed with TAUV to use Titomic’s 3D printing technology to produce Ti alloy rugged soldier uncrewed aerial vehicles (UAVs). The tactical drone market was estimated to be worth AUD 545 million in 2018, according to The Teal Group. Titomic’s first defense drone project demonstrated the advanced manufacturing capabilities of CS technology to produce complex-shaped products with improved performance characteristics. Not only does Titomic Kinetic Fusion make a difference in making metal parts, but it also prints these parts at a considerable scale with a large-scale CS 3D printer capable of printing everything from golf clubs to complex aircraft-wing parts.

At the same time, the literature [63] reported that the General Electric Company (GE) of the United States used CS technology to manufacture completely new parts. For example, gears were fabricated by controlling nozzle movement and external motor drive, and the prepared gears are shown in Figure 9. United Technologies Research Center [74] also produced steel brackets by CS additive manufacturing technology. The research center sprayed the structure on the formed mandrel substrate, removed the substrate, and finally finished the bracket to obtain a steel bracket that met the requirements of use. The finished product is shown in Figure 10.

In 2017, Akron University in Ohio, USA, cooperated with Aviation Maintenance Engineering Services (AMES) to develop CS 3D printing technology for repairing aircraft parts. A related license application was submitted to the Federal Aeronautics Administration (FAA) in the same year. In 2019, Rowan University in New Jersey received $14.5 million from the ARL to develop the Cold Spray Additive Manufacturing (CSAM) program. The materials developed through the research program can be used in the military field so that the military equipment can be lightweight to meet the requirements of use. In 2020, the European Space Agency (ESA) awarded up to GBP 500,000 to researchers in the Department of Mechanical and Manufacturing Engineering at Trinity College Dublin, Ireland, to research CS 3D printing. Nowadays, CS technology is not only used in coating preparation. Due to the unique advantages of this technology, this technology has been applied in the military field abroad. It also has market shares in some civilian fields, such as non-stick pans and printing auto parts. At the same time, commercialization sectors such as civil aviation, automotive, healthcare, and space exploration are also potential customers.

## 5. Interfacial Bond Strength between Cold-Sprayed Coatings and Substrates

The interface bonding strength between the coating and the substrate (adhesive strength) is a crucial factor affecting the overall service performance of the part. To obtain a coating with excellent bonding strength with the substrate, the preferred material for surface repair of parts is the coating material that is consistent with, or close to, the composition of the substrate. However, due to the high activity of Mg alloys, Mg powder is flammable and explosive in the process of CS, so the surface repair of Mg alloy parts is made of Al alloy powder materials [64]. Table 3 [75,76,77,78,79] lists the interfacial bonding strength (tensile and shear strength) and the cohesive strength of the coating material of CS Al-based coatings on the aviation Al alloy and Mg alloy surface, which are compared with the tensile strength of the substrate. It can be seen that the interface bonding strength between the coating and the substrate is much lower than the cohesive force of the coating and the bulk strength of the substrate material, and the coating is easy to peel off at the interface, which seriously affects the integrity of the coating and the substrate during service.

The fundamental reason for the low bond strength between the CS coating and the substrate is the lack of adequate metallurgical bonding at the interface. Studies have revealed that the interfacial bonding between the CS coating and the substrate is characterized by mechanical interlocking and local metallurgical bonding [42,80,81]. To improve the bonding strength of the interface, scholars have conducted much research, among which there are two main methods for improving the effectiveness. One is to generate a diffusion layer between the coating and the substrate by post-annealing. The phase structure of the diffusion layer depends on the reaction characteristics between the coating and the matrix material [82]. Still, the heat treatment will affect the matrix material’s original structure and properties, so it is unsuitable for the treatment of repair parts. The other is to add hard ceramic particles during the coating preparation and use the pinning effect of the hard particles at the coating–substrate interface to improve the bonding strength. However, during the deposition process, the hard ceramic particles significantly impact the surface of the substrate, which quickly leads to the generation of microcracks [79]. Therefore, for repairing aviation Al alloy and Mg alloy parts, scholars are looking for new ways to improve the interface bonding strength between the coating and the substrate to ensure integrity during service.

Through experimental research, Wang and others [39] found that introducing an interfacial bonding transition layer effectively improves the bonding strength between the coating and the substrate. The basic principle of material selection for the bonding transition layer is that it has good chemical activity with both the coating and the substrate and can produce a good interface metallurgical bond with the coating and the substrate during the deposition process of spraying. In widely used aviation Al alloys (such as Al-Zn-Mg-Cu series 7050 and 7075 alloys), Mg alloys (such as Mg-Zn-RE series ZE41 and Mg-Al-Zn series AZ91 alloys), zinc (Zn) is the main alloy component. The solid solubility of Zn in Al varies widely, and the highest solid solubility at the eutectic temperature (381 °C) can reach 83.1% (mass ratio). The crystal structures of Zn and Mg are both close-packed hexagonal structures, and Mg solid solubility of Zn in Mg is 6.2% (mass ratio), and the eutectic temperature is 340 °C. Al-Zn and Zn bonding layers were prepared between the pure Al coating and the interface of 7050 Al alloy and AZ91 Mg alloy substrate, and the bonding strength was significantly improved, as shown in Table 4. The results showed that the interface bonding state between Zn and Al and Zn and Mg can be effectively controlled by adjusting the temperature and speed of the particles during the CS process by using the chemical affinity between Zn, Al, and Mg elements. The interface bonding state between Zn and Al and Zn and Mg can be effectively controlled, a good element diffusion metallurgical bonding layer can be obtained, and the interface bonding strength between the coating and the substrate can be improved. Li et al. [83] reported the preparation of Zn coatings by CS on the surface of mild steel. As shown in Figure 11, with the increase of the spraying gas temperature, the melting degree of the Zn coating surface gradually increases. The melting phenomenon is caused by particle impact (impact-induced melting), which effectively improves the interfacial bonding strength between the Zn coating and the substrate and the cohesion of the Zn coating.

The introduction of texture design on the surface of the substrate can increase the interface contact area and mechanical occlusion between the coating and the substrate and effectively improve the interface bonding strength between the coating and the substrate. Compared with grinding, ion beam, electrical discharge machining, and other methods, laser machining has a series of advantages, such as high machining accuracy, high efficiency, low pollution, small heat-affected zone, and wide application range [84,85]. Kromer et al. [86] studied the influence of the laser processing surface micro-texture on the interfacial bonding strength between CS coating and Al alloy substrate. The study designed textures with two different pore-arrangement densities, as shown in Figure 12. The interfacial bond strength of the microtextured samples can be increased up to 300% (Al-alloy) compared to the samples with the surface treated by sandblasting, as shown in Figure 13. Combined with the fracture morphology analysis, the pore structure of the surface micro-texture not only affects the interface bonding strength between the coating and the substrate, but can also change the fracture mechanism between the coating and the substrate to make it transition from the interface fracture to the internal fracture of the coating, as shown in Figure 14. Therefore, it is of great significance to systematically study the influence of the pore design of the surface micro-texture on the bonding strength and fracture mechanism of the CS coating–substrate interface.

## 6. Preparation of Coatings by Supersonic Laser Deposition Technology

Although the hole pattern design of the surface micro-texture improves the interface bonding strength between the coating and the substrate, the introduction of texture design on the surface of the substrate is cumbersome, and the operation on complex surfaces is difficult. According to the deposition principle of CS technology, the deposition and forming of CS coating are based on the plastic deformation of metal powder materials under high-speed impact, so the powder raw materials need good plasticity [39]. Therefore, it is necessary to find new ways to improve the plasticity of the powder, so that the hard powder can be better deposited and formed, given the shortcomings of CS technology in the process of preparing hard metallic coatings, such as low deposition efficiency, poor coating density, and low bonding strength.

The supersonic laser deposition technology (SLD) is a novel materials-deposition technique which applies a laser source to assist the cold spraying process [87,88,89,90]. The principle of SLD is shown in Figure 15 [78,91]. O’Neill et al. revealed that the simultaneous softening of hard metallic particles by laser can improve the thickness, deposition efficiency, density, and bonding strength of the cold-sprayed coatings. Under the assistance of laser heat, CS can use low-cost nitrogen instead of helium to prepare high-quality coatings with high hardness [90]. Therefore, the cost of coating preparation is reduced, and the selection range of deposition materials is broadened. In recent years, the attraction of SLD technology has increased significantly [92]. Yao et al. [93] used SLD technology and LC technology to prepare hard Ni60 (58–62 HRC) coating on medium carbon steel (AISI 1045 steel) substrate. The results showed that the Ni60 coating prepared by the SLD technology exhibits some characteristics that are better than the coating produced by the traditional LC process, such as fine microstructure, less dilution, and better sliding wear resistance. However, the deposition of hard Ni60 on the steel substrate cannot be achieved by CS technology. Dallin et al. [94] used CS and SLD technology to prepare AISI4340 coating on the surface of the AISI1018 stainless steel substrate. As shown in Figure 16, under the condition of the same spraying process parameters, except for the laser source, the thickness of the CS coating was only 1.7 mm, while the thickness of the SLD coating was 3.5 mm with the deposition rate increase from 48% to 72%. The porosity of the SLD coating was less than 1%. The porosity was found to be slightly higher in the CS deposit than in the SLD deposits with complete bonding between the deposit and the substrate, the coating was denser, and the structure was more uniform. At the same time, due to the laser’s synchronous heating effect, the coating’s hardness was reduced from 561 HV to 466 HV, which effectively alleviated the deformation hardening effect caused by the particle deposition process. Li et al. [95] conducted the solid-state fabrication of continuous and dense WC/Stellite-6 composite coatings by SLD technology. The results of friction and wear tests revealed that the SLD coating had fewer defects and better friction and wear properties compared to the coating prepared by LC, as shown in Figure 17. The bonding mechanism in SLD evolves from predominantly mechanical bonding in CS to co-existence of mechanical and metallurgical bonding. As shown in Figure 18, Yao et al. [78] conducted three-point bending experiments of SLD Stellite6 coating and found that the maximum load of the deposited layer sample was 81.4% that of the cast sample. The fracture morphology showed apparent dimples and tear bands, indicating that the bonding mode between particles was metallurgical bonding. As shown in Figure 19. Gorunov et al. [96,97] prepared 316L stainless steel coatings on stainless steel bars by SLD technology. After testing, the tensile strength of the coating was 650 MPa, and the interface bonding strength between the coating and the substrate could reach 105 MPa.

Table 5 presents the performance comparison of coatings prepared by different technologies. It can be noted that no new phase is produced in the process of SLD coating and CS coating. In contrast, unnecessary phase transformations will occur in LC coating, which will affect the performance and density of the coating. In addition, the average hardness of SLD coating is higher than that of CS coating and LC coating. The friction and corrosion resistance of SLD coatings are better than those of LC coatings, indicating that that SLD technology has certain potential and advantages in the field of coating preparation.

From the research work of the above scholars, it can be seen that SLD technology can effectively overcome the shortcomings of CS technology and effectively improve the interface bonding strength between the coating and the substrate, the strength of the coating itself, and the fatigue performance of the coating. Compared with coating technologies such as thermal spraying and laser cladding, SLD technology inherits the advantages of CS technology, such as high deposition speed, small heat affected zone, and dense coating. SLD technology reduces the critical deposition speed of particles, thereby reducing the gas consumption of the equipment during use, improving the deposition efficiency, and reducing the operating cost of CS technology. According to the above analysis, the future SLD technology will have excellent development prospects and possibilities in preparing high-quality coatings in field of repairing and additive manufacturing of aircraft components.

## 7. Current Challenges and Future Prospects

In summary, compared with traditional coating-preparation technologies, such as flame spraying, arc spraying, and plasma spraying, as well as laser-cladding technology, CS technology can effectively avoid oxidation, phase transformation, thermal cracking, and matrix cracking at the interface caused by thermal influence. At the same time, additive manufacturing through CS technology also has a broad application in rapid production of aircraft parts. SLD technology offsets the shortcomings of CS technology and shows certain advantages in preparing high quality alloy coatings with high hardness. However, CS and SLD technologies still face many challenges in field of aviation application.

(1) CS technology has gradually matured in the additive manufacturing and repair of aviation parts, but there are relatively few types of materials at this stage. More materials need to be tested in future work to form a complete guide map, in terms of coating materials and process parameters.

(2) As an emerging technology, the study of coatings prepared by SLD technology is mainly on the functional properties of friction, wear, and corrosion properties, while there are few studies on the mechanical properties of coatings. In future research, the focus should be placed on the mechanical properties of bonding strength and fatigue performance which are of great importance in aviation applications.

(3) The supersonic powder stream and high-energy laser beams are key players in SLD process. Therefore, it is necessary to use the simulation method combined with experiments to understand the interaction between the kinetic energy and thermal energy conversion of powder materials during the deposition process, in order to precisely control the condition of powder deposition. In addition, the bonding mechanisms of SLD coatings needs to be explored.

## Figures and Tables

**Figure 1 materials-15-08535-f001:**
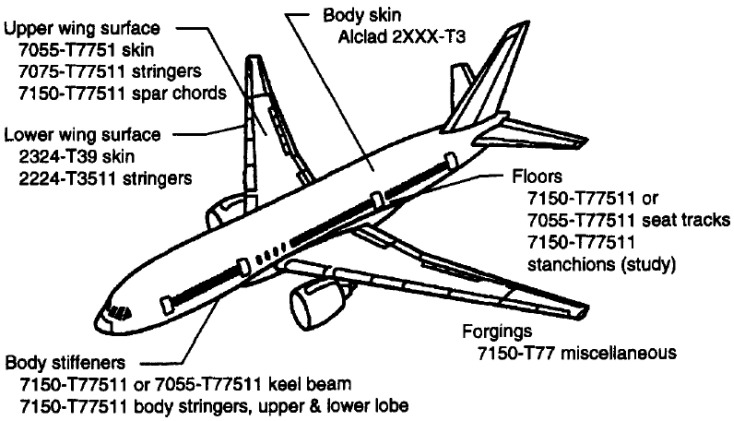
The use of aluminum alloy in Boeing 777 aircraft [9].

**Figure 2 materials-15-08535-f002:**
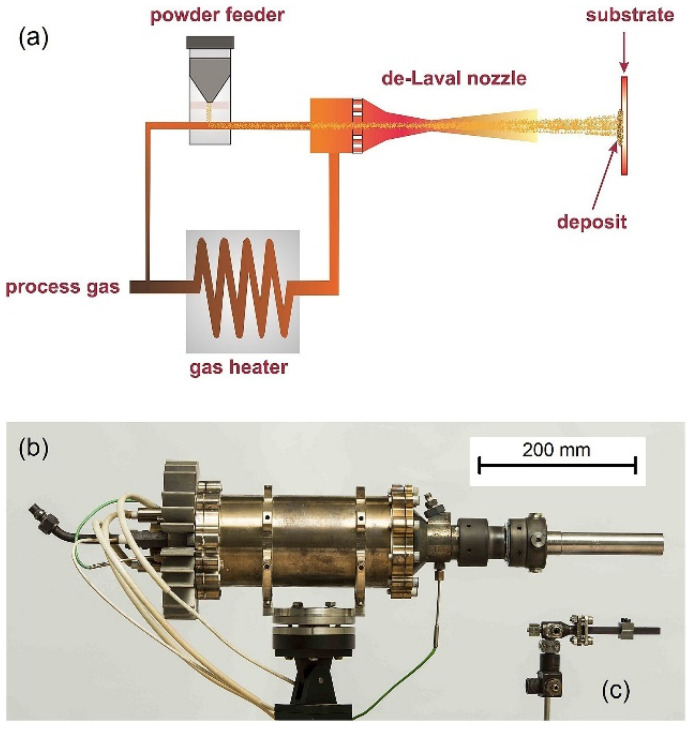
An overview of a high-pressure cold spray system. (**a**) Schematics of the main components of the original system, (**b**) a commercialized cold spray gun/nozzle, and (**c**) an original gun/nozzle [33].

**Figure 3 materials-15-08535-f003:**
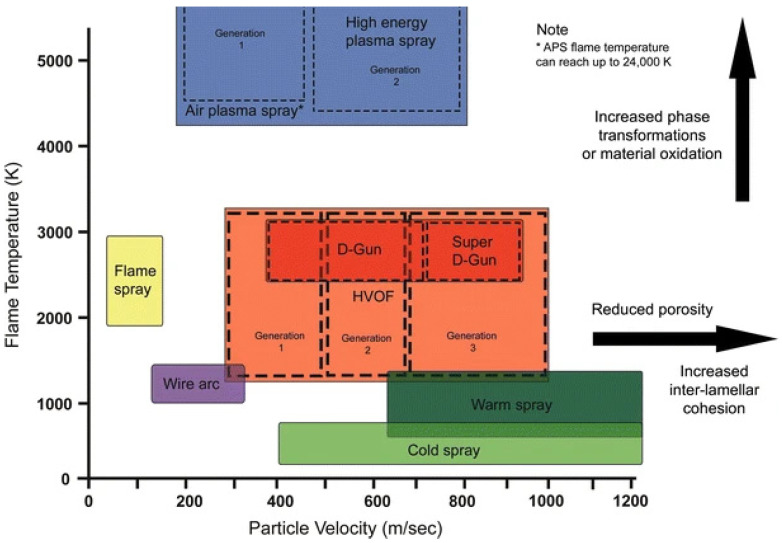
Temperature and velocity profiles of thermal and cold-spray technologies [47].

**Figure 4 materials-15-08535-f004:**
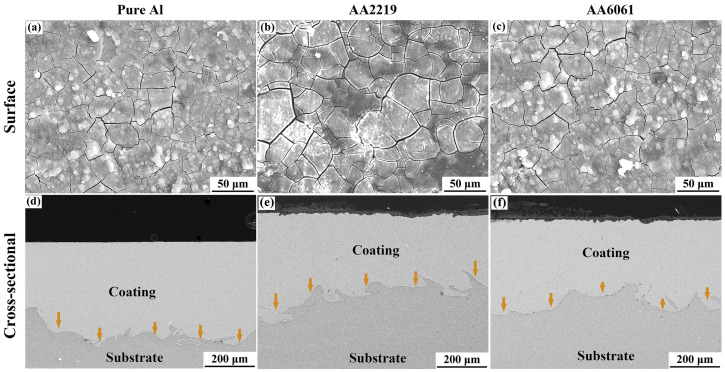
The surface morphologies and cross-sectional microstructures of the MF-CS Al-based coatings after 1000 h immersion. Pure Al (**a**,**d**), AA2219 (**b**,**f**), and AA6061 (**c**,**e**) [48].

**Figure 5 materials-15-08535-f005:**
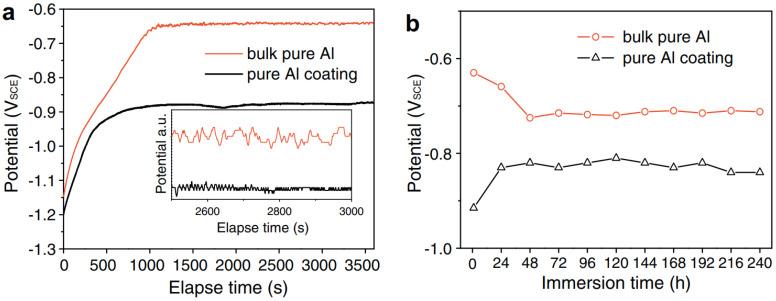
OCP vs. time curves for cold-sprayed pure Al coating and bulk pure Al in 3.5 wt.% NaCl solution in a time span (**a**) from 0 to 1 h (inset: from 2500 to 3000 s) and (**b**) from1 h to 10 days [49].

**Figure 6 materials-15-08535-f006:**
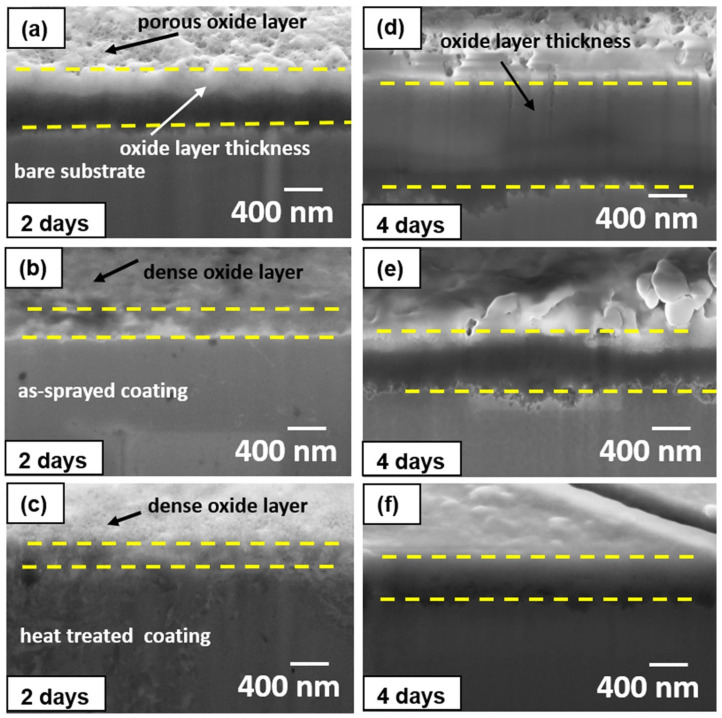
SEM images of milled regions showing the reactive layers after 2 days and 4 days of exposure to salt fog testing of (**a**,**d**) bare substrate, (**b**,**e**) as-sprayed, and (**c**,**f**) heat-treated coatings [52].

**Figure 7 materials-15-08535-f007:**
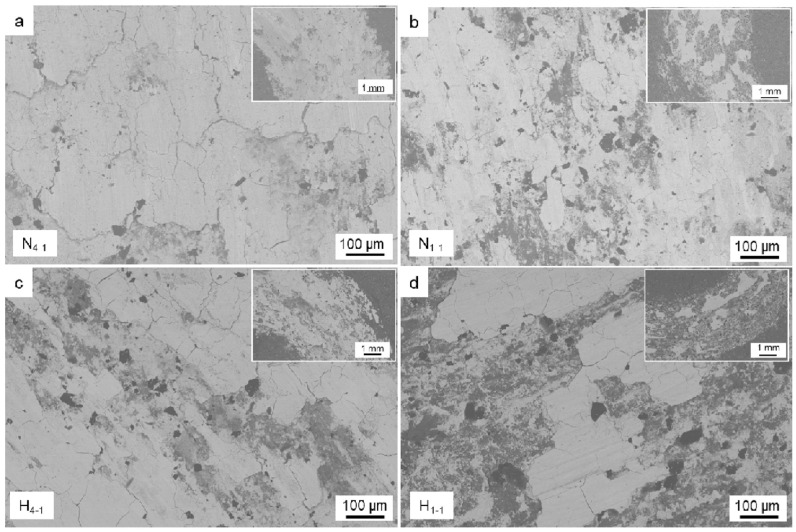
Magnified view of the worn surfaces of the Al/diamond composites under different conditions with the insert image showing the whole worm track [56].

**Figure 8 materials-15-08535-f008:**
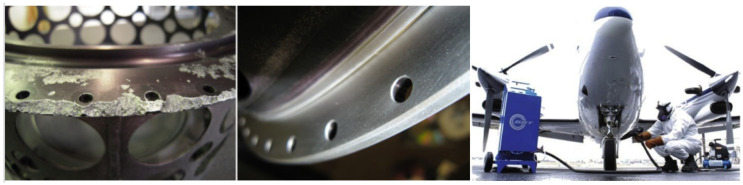
CS technology to repair aviation parts (**left**: before repair, **middle**: after repair) [72] and on-site repair of aircraft landing gear (**right**) [73].

**Figure 9 materials-15-08535-f009:**
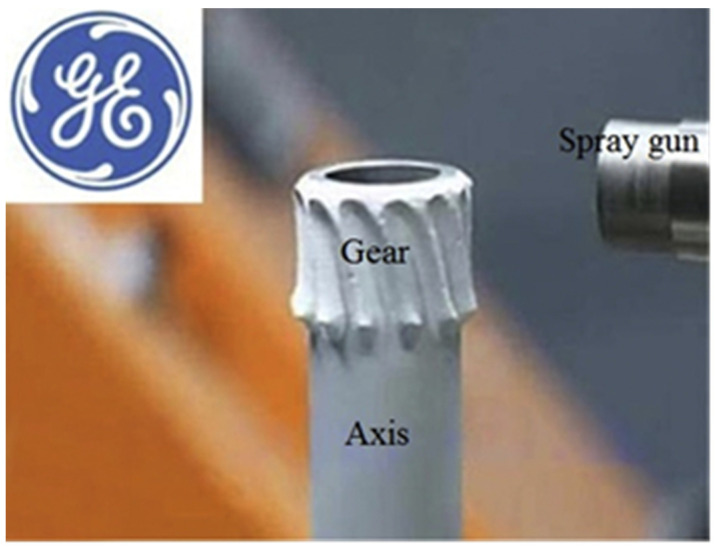
Cold-spray additive manufacturing gear [63].

**Figure 10 materials-15-08535-f010:**
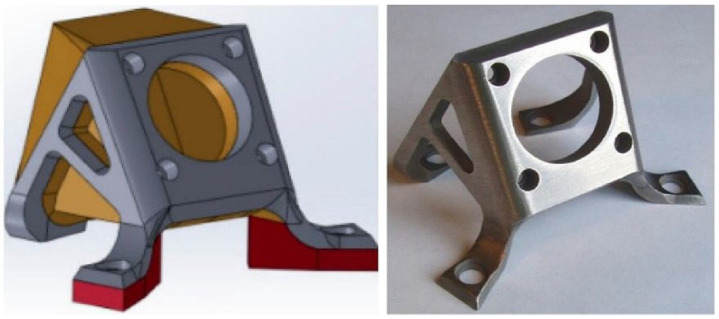
Using cold-spray technology to prepare a steel support [74].

**Figure 11 materials-15-08535-f011:**
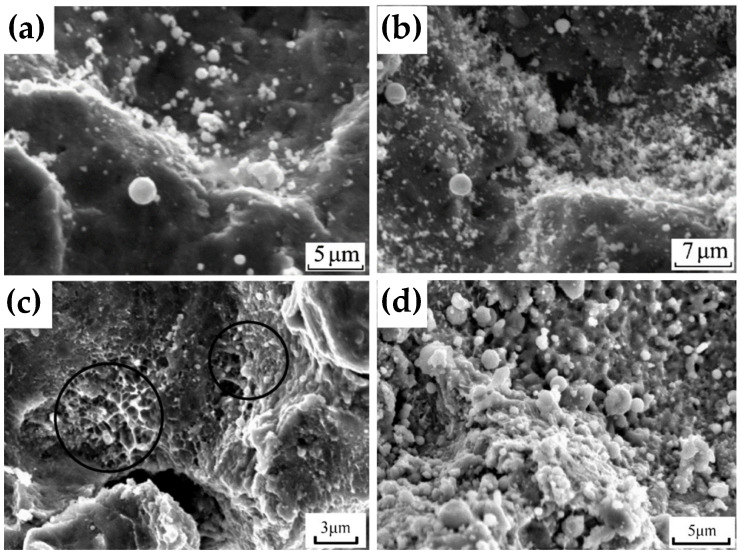
Surface morphology of cold-sprayed zinc coating (spraying gas N_2_) under different spraying gas heating temperatures. (**a**) 165 °C, (**b**) 250 °C, (**c**) 410 °C, and (**d**) 450 °C [83].

**Figure 12 materials-15-08535-f012:**
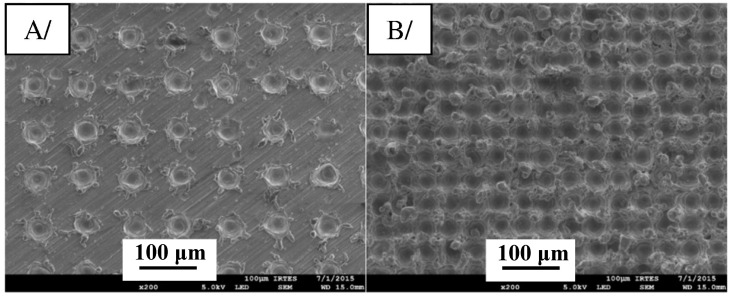
The pore structure of the laser micro-texture on the surface of the cold-sprayed substrate. (**A**) Texture 1 (LST1) and (**B**) texture 2 (LST2) [86].

**Figure 13 materials-15-08535-f013:**
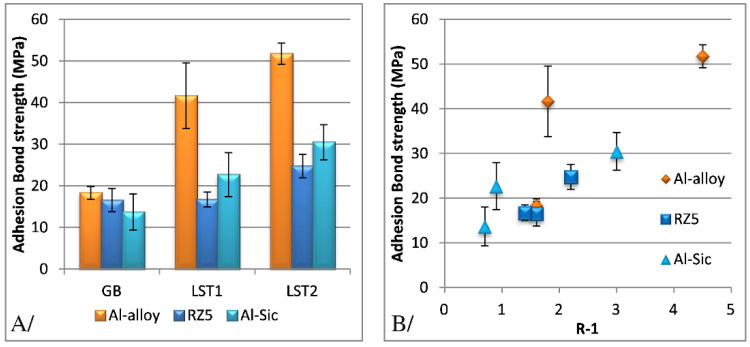
Adhesion bond strength as a function of surface preparation. (**A**) Grit-blasting (GB), texture 1 (LST1), and texture 2 (LST2). (**B**) Adhesion bond strength as a function of the in-contact area ratio for Al, Mg, and Al-SiC couples [86].

**Figure 14 materials-15-08535-f014:**
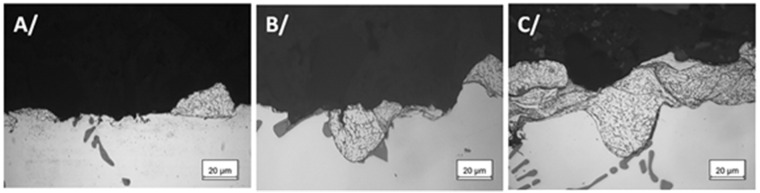
Fracture morphology of the cold-spray coating–substrate interface under different surface conditions. (**A**) GB sandblasting, (**B**) LST1, and (**C**) LST2 [86].

**Figure 15 materials-15-08535-f015:**
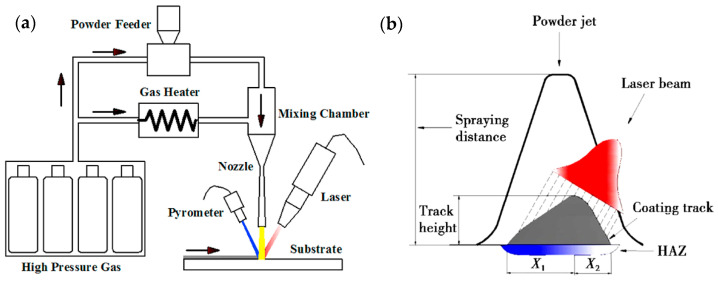
(**a**) Basic structure of supersonic laser deposition system [78] and, (**b**) distribution of laser energy and powder deposition state in deposition area [87].

**Figure 16 materials-15-08535-f016:**
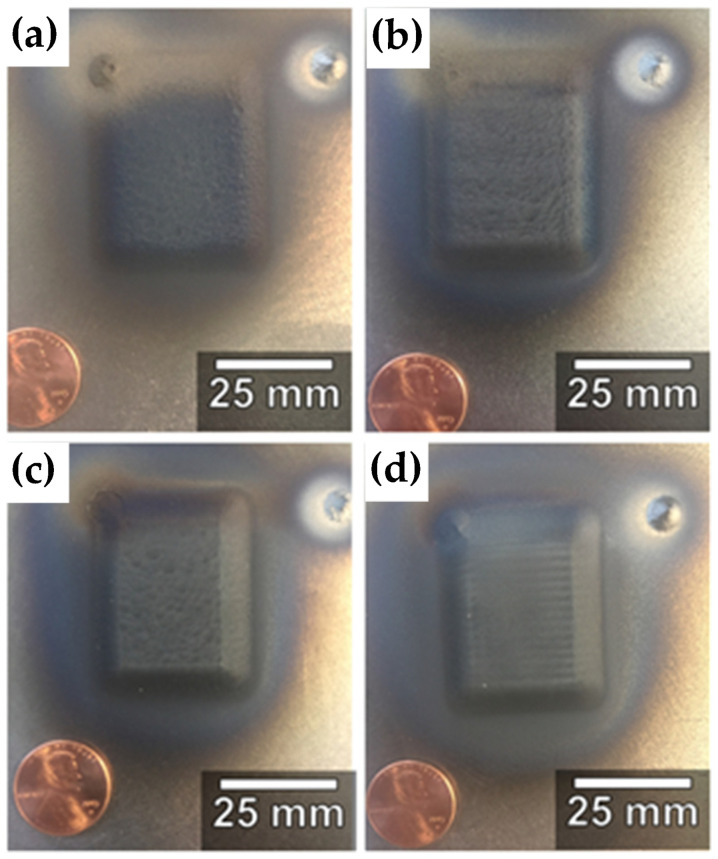
Photographs of (**a**) CS (no laser) and LACS deposits at surface temperatures of (**b**) 500 °C, (**c**) 738 °C, and (**d**) 950 °C. A copper penny is placed next to the deposit for size reference [94].

**Figure 17 materials-15-08535-f017:**
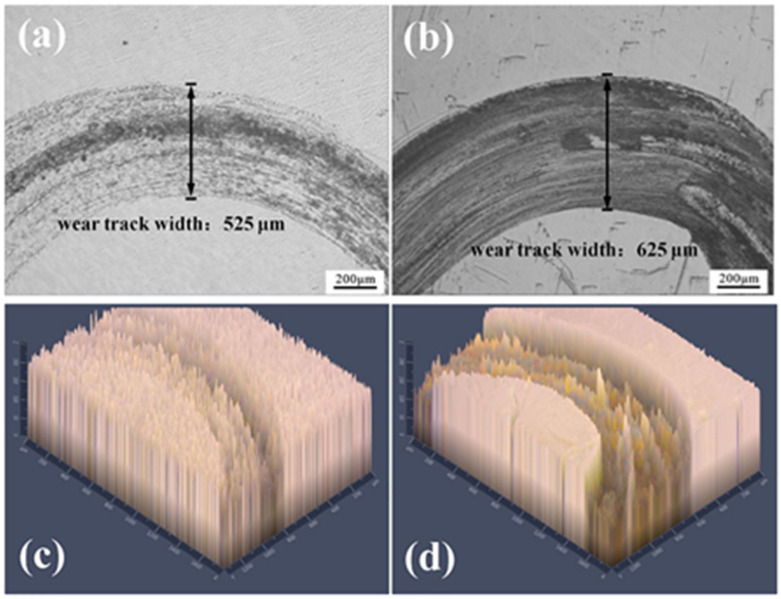
Three-dimensional microscopic analysis of the worn surfaces of the SLD coating (**a**,**c**) and the LC coating (**b**,**d**) SLD coating [95].

**Figure 18 materials-15-08535-f018:**
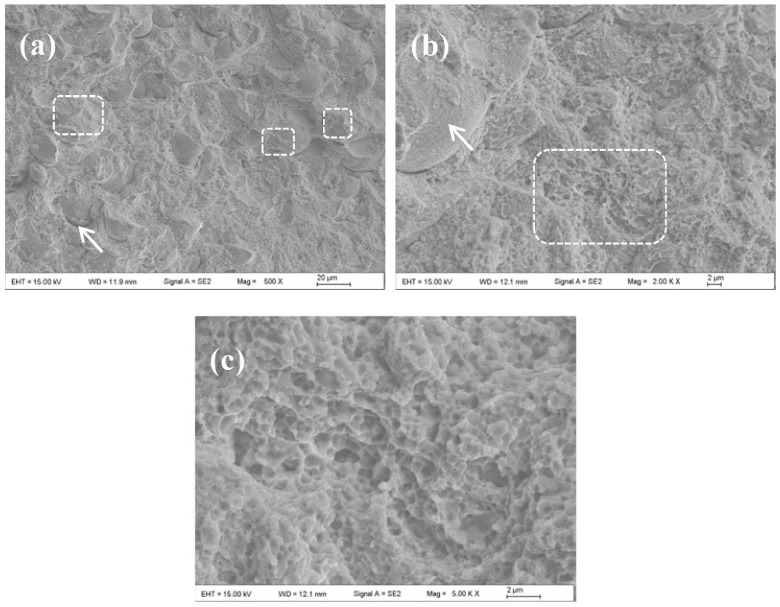
SEM morphologies of fractured surface of the SLD Stellite 6 coating specimen (**a**,**b**) spherical particle surfaces with tearing edges as marked by boxes and arrows. (**c**) the first morphology suggests the rupture at particle–particle interface [78].

**Figure 19 materials-15-08535-f019:**
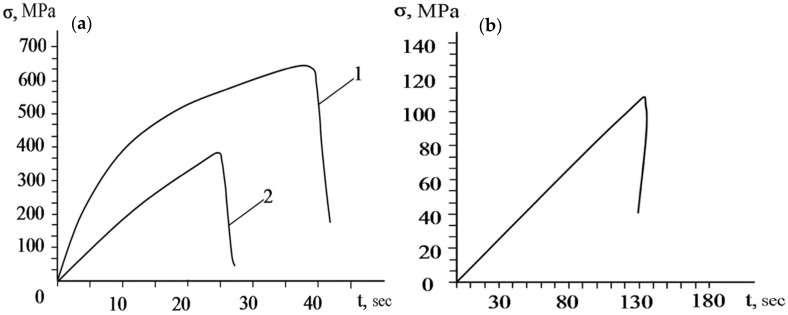
(**a**) Diagram of tensile test specimens with the adhesive strength under shear load and (**b**) curves of tensile specimens cut from the coating produced by supersonic laser deposition in the longitudinal direction (1) and in the transverse direction (2) [96].

**Table 1 materials-15-08535-t001:** Structural material usage in large commercial aircraft [9].

Aircraft	Aluminum	Steel	1Titanium	PMCs	Other
Boeing747	81 wt.%	13 wt.%	4 wt.%	1 wt.%	1 wt.%
Boeing757	78 wt.%	12 wt.%	6 wt.%	3 wt.%	1 wt.%
Boeing767	80 wt.%	14 wt.%	2 wt.%	3 wt.%	1 wt.%
Boeing777	70 wt.%	11 wt.%	7 wt.%	11 wt.%	1 wt.%
Boeing787	20 wt.%	10 wt.%	15 wt.%	50 wt.%	5 wt.%
Airbus A380	61 wt.%	10 wt.%	10 wt.%	22 wt.%	7 wt.%
Airbus A350	20 wt.%	7 wt.%	14 wt.%	52 wt.%	7 wt.%
DC-10	78 wt.%	14 wt.%	5 wt.%	1 wt.%	2 wt.%
MD-11	76 wt.%	9 wt.%	5 wt.%	8 wt.%	2 wt.%
MD-12	70 wt.%	8 wt.%	4 wt.%	16 wt.%	2 wt.%

**Table 2 materials-15-08535-t002:** Comparison of different technologies for preparing alloy coatings on the surface.

Coating Material		Coating Technology	Performance	Literature
Fe-based	MoS_2_/Fe-based	HVOF	Friction and wear	[25]
	FeCrB + Al FeCr + Al + C	Arc spray	Friction and wear	[18]
	Fe-based amorphous coatings	HVOF	Corrosive property	[26]
Al-based	NiAl nano powders	HVOF	Friction and wear	[16]
	Al-based amorphous–nanocrystalline composite coatings	Laser cladding	Mechanical properties and corrosion performance	[27]
NiCr-based	NiCrB (B:0–4 wt.%)	Wire-arc spray	Hot corrosion performance	[19]
Cu-based	MoS_2_/Cu-Al	Thermal spray	Friction and wear	[28]

**Table 3 materials-15-08535-t003:** Mechanical properties of cold-spray coatings on aluminum and magnesium alloys.

Serial Number	Substrate Material		Substrate Tensile Strength (MPa)	Coating Material	Coating Cohesion (MPa)	Coating–Substrate Bond Strength		Literature
						Tensile Adhesion Strength (MPa)	Shear Adhesion Strength (MPa)	
1	Al alloy	AA2024-T35	420	CP-Al	-	43	66	[75]
2		6061-T6	240	6061	322	78	73	[76]
3		7075	525	7075	368	84	92	[77]
4	Mg alloy	ZE41A	115	CP-Al	-	71	-	[78]
5		AZ91D	160	Al + Al_2_O_3_	110	-	45	[79]

**Table 4 materials-15-08535-t004:** Matrix-coating interface bonding strength [37].

Serial Number	Substrate Material	Bonding-Layer Material	Coating Material	Coating–Substrate Tensile Strength (MPa)
1	7050 Al alloy	-	CP-Al	62
2	7050 Al alloy	Al-15 wt.%Zn	CP-Al	72
3	AZ91 Mg alloy	-	CP-Al	34
4	AZ91 Mg alloy	Zn	CP-Al	57

**Table 5 materials-15-08535-t005:** Performance comparison of coatings prepared by different technologies.

Coating Material	Coating Technology	Phase Composition	Average Hardness	Other	Literature
AISI4340	Cold spray	Without new phase	561 Hv	Deposition efficiency: 48%	[94]
	Supersonic laser deposition	Without new phase	592 Hv	Deposition efficiency: 70%	
WC/Stellite-6	Laser cladding	New phase generation	851 HV_50_	Friction coefficient: 0.75	[95]
	Supersonic laser deposition	Without new phase	893 HV_50_	Friction coefficient: 0.62	
Stellite-6	Laser cladding	New phase generation	nano-hardness: 5.42 ± 1.33 GPa	Mass loss rate: 0.045 mg min^−1^	[98]
	Supersonic laser deposition	Without new phase	nano-hardness: 8.91 ± 0.27 GPa	Mass loss rate: 0.01 mg min^−1^	
Ni60	Laser cladding	New phase generation	625 ± 55 HV_0.3_	Friction coefficient: 0.82	[93]
	supersonic laser deposition	Without new phase	867 ± 24 HV_0.3_	Friction coefficient: 0.68	

## Data Availability

Not applicable.

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
