# Peer review of "Research Progress of Coating Preparation on Light Alloys in Aviation Field: A Review"

_materials, 2022, doi:10.3390/ma15238535_

Round 1

Reviewer 1 Report

The presented manuscript includes the study of the application of research progress of coating preparation on light alloys in aviation field: a review. 

Some general corrections and weak points should be mentioned. 

1. It would be great to show in the review paper more exact information. E.g., complete the phrase “include Fe-based, Cu-based, Al-based, and other alloy coatings” with exact compositions. Please, add more compositions through the text.

2. Would be also very useful to provide a table with a comparative analysis of the different coating preparation approaches by some chosen parameters. 

Author Response

The corrections/changes marked on annotated manuscript have been revised and highlighted.

Point 1: It would be great to show in the review paper more exact information. E.g., complete the phrase “include Fe-based, Cu-based, Al-based, and other alloy coatings” with exact compositions. Please, add more compositions through the text.

Response 1: The authors agree with the reviewer. Replacement and modification have been completed in the manuscript, as seen in Table 2.

Table 2. Comparison of different technologies for preparing alloy coatings on the Surface.

Coating Material

Coating technology

performance

Literature

Fe-based

MoS2/Fe-based

HVOF

Friction and wear

[25]

FeCrB+Al

FeCr+Al+C

arc spray

Friction and wear

[18]

Fe-based amorphous coatings

HVOF

Corrosive performance

[26]

Al-based

NiAl nano powders

HVOF

Friction and wear

[16]

Al-based amorphous-nanocrystalline composite coatings

Laser cladding

Mechanical properties and corrosion performance

[29]

NiCr-based

NiCrB (B:0–4 wt%)

wire-arc spray

Hot corrosion performance

[19]

Cu-based

MoS2/Cu-Al

Thermal spray

Friction and wear

[27]

Point 2: Would be also very useful to provide a table with a comparative analysis of the different coating preparation approaches by some chosen parameters.

Response 2: The authors agree with the reviewer. Replacement and modification have been completed in the manuscript, “In the last part, the author added tables and descriptions for the performance comparison of different coating technologies.

Table 5 presents the performance comparison of coatings prepared by different technologies. It can be noted that no new phase is produced in the process of SLD coating and CS coating. In contrast, unnecessary phase transformations will occur in LC coating, which will affect the performance and density of the coating. In addition, the average hardness of SLD coating is higher than that of CS coating and LC coating. The friction and corrosion resistance of SLD coatings are better than those of LC coatings, indicating that that SLD technology has certain potential and advantages in the field of coating preparation.

Table 5. the performance comparison of coatings prepared by different technologies.

Coating Material

Coating technology

Phase composition

Average hardness

other

Literature

AISI4340

Cold spray

without new phase

561Hv

Deposition efficiency: 48%

[94]

supersonic laser deposition

without new phase

592Hv

Deposition efficiency: 70%

WC/Stellite-6

Laser cladding

new phase generation

851HV50

friction coefficient: 0.75

[95]

supersonic laser deposition

without new phase

893HV50

friction coefficient: 0.62

Stellite-6

Laser cladding

new phase generation

nano-hardness: 5.42±1.33GPa

mass loss rate:0.045mg min−1

[98]

supersonic laser deposition

without new phase

nano-hardness: 8.91±0.27GPa

mass loss rate:0.01mg min−1

Ni60

Laser cladding

new phase generation

625±55 HV0.3

friction coefficient: 0.82

[93]

supersonic laser deposition

without new phase

867±24 HV0.3

friction coefficient: 0.68

Reviewer 2 Report

This Manuscript should add a paragraph on the histroy of Cold Spray Techonology in first chapter. It was originated by Anatply Papyrin ,et al. in the 1980s.  I wnat you to respect the originator.

It's need to add the chapter about the comparison of deposition technologies and properties of films of CS, SLD and conventional methods in the first half.

 I can't read the comment of Fig.3. you need to redraw cleary.

Please consider this matter.

Author Response

The corrections/changes marked on annotated manuscript have been revised and highlighted.

Point 1: This Manuscript should add a paragraph on the history of Cold Spray Technology in first chapter. It was originated by Anatply Papyrin et al. in the 1980s.  I want you to respect the originator.

Response 1: The authors agree with the reviewer. Replacement and modification have been completed in the manuscript, i.e., “In the mid-1980s, when the Soviet Union scientist Anatolii Papyrin and his team studied the model of two-phase flow (gas + solid particles) in a wind tunnel, they found that when the speed of the solid particles was higher than a certain critical value, the matrix material would no longer be eroded and particles began to adhere to the matrix [32]. Inspired by this phenomenon, Anatolii Papyrin and his team first proposed the concept of cold gas dynamic spraying (referred to as ‘cold spraying’) in 1990, which could be used as a new coating process. Later, they successfully deposited a variety of pure metals, metal alloys and composites on different matrix, proving the possibility of cold spraying technology in engineering applications [32].

[32] Papyrin A. Cold spray technology. Adv. Mater. Process. 2001, 159, 49–51.

Point 2: It's need to add the chapter about the comparison of deposition technologies and properties of films of CS, SLD and conventional methods in the first half.

Response 2: The authors agree with the reviewer. Replacement and modification have been completed in the manuscript, “In the last part, the author added tables and descriptions for the performance comparison of different coating technologies.

Table 5 presents the performance comparison of coatings prepared by different technologies. It can be noted that no new phase is produced in the process of SLD coating and CS coating. In contrast, Unnecessary phase transition will occur in laser cladding coating. The average hardness of SLD coating is higher than that of CS coating and laser cladding coating. The friction and corrosion resistance of SLD coatings are better than those of laser cladding coatings, indicating that SLD technology has certain potential and advantages in the field of coating preparation.

Table 5. the performance comparison of coatings prepared by different technologies.

Coating Material

Coating technology

Phase composition

Average hardness

other

Literature

AISI4340

Cold spray

without new phase

561Hv

Deposition efficiency: 48%

[94]

supersonic laser deposition

without new phase

592Hv

Deposition efficiency: 70%

WC/Stellite-6

Laser cladding

new phase generation

851HV50

friction coefficient: 0.75

[95]

supersonic laser deposition

without new phase

893HV50

friction coefficient: 0.62

Stellite-6

Laser cladding

new phase generation

nano-hardness: 5.42±1.33GPa

mass loss rate:0.045mg min−1

[98]

supersonic laser deposition

without new phase

nano-hardness: 8.91±0.27GPa

mass loss rate:0.01mg min−1

Ni60

Laser cladding

new phase generation

625±55 HV0.3

friction coefficient: 0.82

[93]

supersonic laser deposition

without new phase

867±24 HV0.3

friction coefficient: 0.68

Point 3: I can't read the comment of Fig.3. you need to redraw clearly.

Response 3: The authors agree with the reviewer. Replacement and modification have been completed in the manuscript, i.e., Figure 3. Temperature and velocity profiles of thermal and cold spray technologies.

Round 2

Reviewer 1 Report

paper can be suggested for publishing